

# Subclinical hypothyroidism during pregnancy and the impact of levothyroxine therapy on pregnancy outcomes in women

Yutian Zhou[1,2,*], Yi Wang[3,*], Tianxiao Yu[1], Yuan Li[4], Meiyan Mi[5], Jianqiang Su[4] and Jun Ge[1,2]

[1] Research Center for Clinical Medical Sciences, The Fourth Hospital of Shijiazhuang, Shijiazhuang, China
[2] School of Public Health, Hebei Medical University, Shijiazhuang, China
[3] Department of Clinical Laboratory, The Fourth Hospital of Shijiazhuang, Shijiazhuang, China
[4] Department of Medicine, The Fourth Hospital of Shijiazhuang, Shijiazhuang, China
[5] Department of Gynecology, The Fourth Hospital of Shijiazhuang, Shijiazhuang, China
[*] These authors contributed equally to this work.

Corresponding author
Jun Ge, GJ8005AA@163.com

## ABSTRACT

**Aims.** The purpose of this study is to investigate the impact of subclinical hypothyroidism (SCH) during pregnancy and levothyroxine (LT4) therapy on pregnancy outcomes.

**Methods.** Among 6,510 pregnant women who came to The Fourth Hospital of Shijiazhuang for pregnancy examination and delivery, 266 pregnant women with SCH and treated with LT4 were selected as the SCH group and 672 pregnant women without SCH were selected as the non-SCH group, and the incidence rates of adverse pregnancy outcomes in pregnant women and newborns of the two groups were compared using Chi-square test and logistic regression. According to the therapeutic effect, pregnant women treated with LT4 were categorized into sustained euthyroid status (SES) and suboptimal thyroid status (STS) groups and compared with the non-SCH group using chi-square test. The correlation of thyroid stimulating hormone (TSH) levels at different stages of pregnancy was explored using Spearman's rank test.

**Results.** The incidence of hypertensive disorders of pregnancy (HDP), premature rupture of membranes (PROM), and neonatal outcomes were ventricular or atrial septal defect (V/ASD), hyperbilirubinemia, and pneumonia were higher in the SCH group (SCH pregnant women) than in the non-SCH group (non-SCH pregnant women) ($p < 0.05$). The incidence of multiple maternal and neonatal complications was higher in the SCH-STS group (SCH in two or three gestational trimesters) compared to the SCH group. With a tendency for TSH levels to increase as the pregnancy progressed.

**Conclusion.** SCH during pregnancy is associated with a high incidence of various pregnancy complications, and LT4 therapy that controls serum TSH levels at normal levels throughout pregnancy can reduce these risks.

## INTRODUCTION

Thyroid disease is one of the most common endocrine disorders in clinical practice, with the highest prevalence in women, especially those of childbearing age (*Wu et al., 2015*). Subclinical hypothyroidism (SCH) is a disorder defined as the finding of elevated serum thyroid stimulating hormone (TSH) levels but normal free thyroxine (FT4) levels, and is a common clinical condition during pregnancy. If left untreated, it may lead to progressive thyroid dysfunction, which can affect normal pregnancy by endangering maternal and neonatal health (*Hammond et al., 2015*). Regarding the harms of SCH, numerous studies have shown that untreated patients with SCH have a higher risk of one or more adverse pregnancy outcomes (pregnancy loss, preterm birth (PTB), hypertension and low birth weight (LBW)) compared to pregnant women with normal thyroid function (*Negro et al., 2010*; *Casey et al., 2005*; *Wilson et al., 2012*; *Schneuer et al., 2012*; *Chen et al., 2014*). A retrospective cohort study of Chinese women showed that SCH during pregnancy increased the risk of hypertensive disorders of pregnancy (HDP), especially among women diagnosed with the disease in the first and second trimesters (*Wu et al., 2019*). However, some other studies have not found any association between SCH and pregnancy complications (*Männistö et al., 2010*; *Männistö et al., 2009*; *Sahu et al., 2010*; *Cleary-Goldman et al., 2008*).

Due to ethnic and regional differences, the American Thyroid Association (ATA) recommends that each regional unit establish a specific reference range for serum TSH levels during pregnancy. If not, it is recommended that the upper limit of the reference value for serum TSH levels in the first trimester of pregnancy be established at 4.0 mIU/L (*Alexander et al., 2017*). Levothyroxine (LT4) therapy is currently commonly used to treat SCH during pregnancy. The aim is to keep serum TSH levels within the normal range (*De Groot et al., 2012*; *Shan & Teng, 2019*). Nevertheless, there is a lack of supporting evidence for the benefit of LT4 therapy in improving health status of pregnant women with SCH. A 2016 meta-analysis on SCH in pregnancy showed that there was insufficient evidence to support that patients with SCH in pregnancy can benefit from LT4 therapy (*Sankoda et al., 2024*). A study in 2022 showed that benefit for pregnant women was related to early initiation of the treatment in pregnancy (*Dash et al., 2022*).

Although universal thyroid function screening during pregnancy remains controversial in other countries, as of 2022, China has 19 free pre-conception and pregnancy examinations that include thyroid function tests. In the absence of universal pre-conception or early pregnancy thyroid testing, previous studies on the benefits of LT4 therapy on pregnancy outcomes in SCH patients have generally reported those outcomes in the setting of starting LT4 therapy towards the end of the first trimester (*Shan & Teng, 2019*). In that setting, SCH status in pregnant women may not be promptly detected, and the effect of LT4 therapy may be diminished. Furthermore, we found that previous studies have tended to focus on a short period after LT4 treatment and have not addressed the entire subsequent gestation period, the thyroid function status of pregnant women across much of the pregnancy may have been incorrectly assessed and erroneously assumed to have been normal.

The results of a 2018 survey showed a significant increase in the prevalence of SCH in China over the past decade or so, with a similar increase in the thyroid peroxidase (TPO)-Ab-positive people (*Taylor et al., 2018*). The disease burden associated with subclinical hypothyroidism is increasing. In this context, a retrospective cohort study was conducted to investigate the impact of SCH in pregnancy on maternal and neonatal pregnancy outcomes and to explore the benefits of timely LT4 therapy for SCH on pregnancy outcomes. This study will provide a basis for the management of SCH in pregnancy.

## MATERIALS AND METHODS

### Research object

From January 2019 to December 2020, a total of 938 pregnant women who underwent obstetric examinations and deliveries at the The Fourth Hospital of Shijiazhuang were invited to participate in this study. We applied the following inclusion criteria: (1) age 18–45; (2) no family history of psychiatric disorders; (3) data of thyroid function tests in three pregnancy trimesters. The exclusion criteria were as follows: (1) pre-pregnancy diabetes or hypertension; (2) all types of abortion; (3) history of any type of thyroid disease and LT4 treatment; (4) twin or multiple, in vitro fertilization and embryo transfer (IVF-ET) pregnancies, any pregnancy assisted by medication; (5) birth history of stillbirths or major malformations. Subjects were treated with LT4 according to the doctor immediately when they were first diagnosed with SCH, then the pregnant women were tested for thyroid function indicators every three to four weeks of pregnancy to confirm that the dose was appropriate or if the dose needed to be adjusted. Diagnostic criteria for SCH in pregnancy: serum TSH level > 4.0 mIU/L, FT4 level: 12–22 pmol/L (*Endocrinology Branch of Chinese Medical Association, Perinatal Medicine Branch Chinese Medical Association, 2019*). The serum test of thyroid function was performed by radioimmunotherapy in our laboratory. Ultimately, out of the population of 6510 pregnant women, 672 women without SCH in pregnancy were included in the non-SCH group and 266 women with SCH in pregnancy were included in the SCH group. Forty-seven of the study subjects had lab results indicative of SCH in at least two trimesters of pregnancy and were included in the STS group: 12 pregnant women with SCH during the first and second trimesters, 11 pregnant women with SCH during the first and third trimesters, 16 pregnant women with SCH during the second and third trimesters, eight pregnant women with SCH during all three stages of pregnancy. All the participants were informed about the study and a signed permission document was obtained in accordance with the Declaration of Helsinki. The Ethics Committee of The Fourth Hospital of Shijiazhuang approved the study (NO. 20230109). Figure 1 showed the details.

### Data collection

Data were collected as previously described in *Lingli et al. (2022)*. The following questions were asked of the study participants through an electronic questionnaire. The questions included: maternal age, height, pre-pregnancy weight, education, parity, family income, family history of diabetes, smoking or exposure to secondhand smoke, alcohol consumption during pregnancy, folic acid, vitamin supplementation and so on.
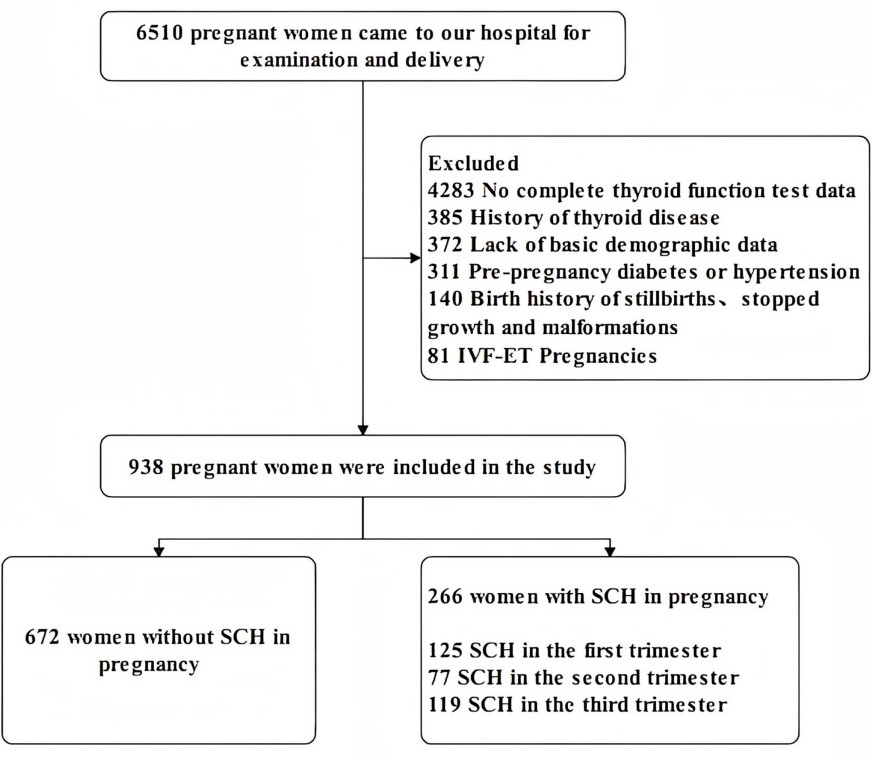

**Figure 1** Flow chart of participants participation in the study.

Adverse pregnancy outcome collection: pregnant women: gestational diabetes mellitus (GDM); HDP (including pre-eclampsia and eclampsia); premature rupture of membranes (PROM). Newborn: premature birth (PTB); abnormal weight birth (AWB), including low birth weight (LBW) and large babies; hyperbilirubinemia and other adverse neonatal events (RDS).

GDM was diagnosed based on the results of the 75 g oral glucose tolerance test (OGTT) at 24–28 weeks of gestation when the blood glucose met at least one of the following criteria: fasting plasma glucose (FPG) 5.1–6.9 mmol/L, 1 hr plasma glucose (PG1H) $\geq$10 mmol/L, or 2 hr plasma glucose (PG2H) 8.5–11 mmol/L. PTB is defined as live births prior to the completion of 37 weeks of gestation. AWB is defined as a newborn weighing more than 4,000 g, or less than 2,500 g. The diagnostic criteria for other diseases are mainly based on the 2008 People's Health Press *Chinese Obstetrics and Gynaecology* (*Cao, 2008*).

## Statistical analysis

SPSS 22.0 statistical software was used to analyse the data. For measured data, the mean $\pm$ standard deviation was used to describe the data, and the count data were described as the number of cases (%). The $\chi^2$-test or Fisher's exact test were used to compare pregnancy outcomes between the different groups. The association between subclinical hypothyroidism in pregnancy and adverse pregnancy outcomes was further examined using a multifactorial logistic regression model, after adjusting for confounding

factors such as age and pre-pregnancy body mass index (BMI). Spearman's correlation analysis was used to assess the relationship between TSH levels at different trimesters. All statistical tests were performed using a two-sided test, and $p$-value $< 0.05$ was considered statistically significant.

## RESULTS

Table 1 describes the baseline characteristics of the participants. There was no statistically significant difference in any of the characteristic indicators between the SCH and non-SCH groups ($p > 0.05$).

Table 2 shows the occurrence of maternal and neonatal pregnancy outcomes in the SCH and non-SCH groups. In terms of pregnant women, the SCH group had a higher prevalence of HDP and PROM ($p < 0.05$). In terms of newborns, the proportion of newborns with ventricular or atrial septal defect (V/ASD), hyperbilirubinemia, and pneumonia was higher in the SCH group ($p < 0.05$). For other outcomes, no significant differences were found between the two groups.

Compared with non-SCH pregnant women in the same period, pregnant women with SCH in the first trimester had higher rates of GDM, HDP, neonatal pneumonia. Pregnant women with SCH in the second trimester had higher rates of HDP, PTB, AWB, hyperbilirubinemia and RDS. Pregnant women with SCH in the third trimester have higher rates of PTB ($p < 0.05$). Table 3 showed the details.

Table 4 shows the results of a multifactorial logistic regression of SCH in pregnancy and five adverse pregnancy outcomes. The results of model 3, which was adjusted for other factors such as age and BMI, showed that SCH in pregnancy was associated with a higher risk of hyperbilirubinemia and neonatal pneumonia.

Table 5 shows the comparison of pregnancy outcomes between the sustained euthyroid status (SES), suboptimal thyroid status (STS) and non-SCH group of SCH during pregnancy. The attack rate of HDP, PROM, PTB, AWB, hyperbilirubinemia, V/ASD, and RDS was higher in the uncured group as compared to the non-SCH group, whereas there was no statistically significant difference between the SES group and the non-SCH group.

Table 6 shows the results of Spearman's rank correlation of TSH levels in different trimesters and in the entire population, TSH levels in the latter trimester were positively correlated with TSH levels in the previous trimester. In the non-SCH group, the population without SCH showed the same results. The highest correlation was found between TSH levels in the second and third trimester.

## DISCUSSION

A single-center study was designed to evaluate the impact of SCH after LT4 treatment on pregnancy outcomes and the benefits of LT4 treatment for SCH on pregnancy outcomes. The diagnostic criteria for SCH in this study are in accordance with the Chinese Guidelines for the Management of Thyroid Disorders During Pregnancy and Childbirth (*Endocrinology Branch of Chinese Medical Association, Perinatal Medicine*

**Table 1  Basic characteristics of pregnant women.**

| Characteristic | | SCH group $n = 266$ | Non-SCH group $n = 672$ | $p$ |
|---|---|---|---|---|
| Age | Years | 29.24 ± 3.74 | 28.81 ± 3.54 | 0.102 |
| BMI | kg/m$^2$ | 21.63 ± 3.74 | 21.66 ± 2.84 | 0.111 |
| | Low | 22 (13.40) | 79 (12.80) | |
| Education | Medium | 62 (37.80) | 249 (40.36) | 0.838 |
| | High | 80 (48.80) | 289 (46.84) | |
| Smoking | No | 233 (87.60) | 591 (87.90) | 0.882 |
| | Yes | 33 (12.40) | 81 (12.10) | |
| Alcohol consumption | No | 256 (96.20) | 657 (97.80) | 0.191 |
| | Yes | 10 (3.80) | 15 (2.20) | |
| Family history of diabetes | No | 246 (92.50) | 629 (93.6) | 0.537 |
| | Yes | 20 (7.50) | 43 (6.40) | |
| Family income | Low | 80 (30.1) | 222 (33.0) | |
| | Medium | 110 (41.40) | 230 (34.20) | 0.234 |
| | High | 76 (28.60) | 220 (32.70) | |
| Parity | One | 174 (65.40) | 470 (69.90) | |
| | Two | 81 (30.50) | 182 (27.10) | 0.347 |
| | Three | 11 (4.10) | 20 (3.00) | |
| Folic acid supplementation | Zero | 21 (7.90) | 30 (4.50) | |
| | Pre-pregnancy | 105 (39.50) | 273 (40.60) | 0.112 |
| | First trimester | 140 (52.60) | 369 (54.90) | |
| VD supplementation | No | 107 (40.20) | 269 (40.00) | |
| | Pre-pregnancy | 53 (19.90) | 107 (15.90) | 0.276 |
| | First trimester | 106 (39.80) | 296 (44.00) | |

**Notes.**

High school graduation and below is defined as low education, university graduation or above is defined as high education; pregnant women who smoke or are exposed to second-hand smoke for more than fifteen minutes per week are defined as smoking; annual family income below RMB100,000 is defined as low income and above RMB200,000 is defined as high income.

*Branch Chinese Medical Association, 2019*). In a study conducted in Jiangxi, China, among 2,636 Chinese local inhabitants, 14.4%, 44.5%, 26.1%, and 15.0% had deficient, adequate, more than adequate, and excessive iodine concentrations, respectively. The prevalence rates of hyperthyroidism, subclinical hyperthyroidism, hypothyroidism, subclinical hypothyroidism, thyroid nodules, and thyroid autoimmunity were 0.91%, 0.57%, 0.34% and 7.89%, 9.45%, and 12.7%, respectively (*Yan et al., 2023*). It is well known that iodine intake is closely related to thyroid diseases, and the incidence rate of thyroid diseases in China should be of great concerns.

The study found that all participants received LT4 treatment after diagnosis of SCH in pregnancy. However, there was an increased incidence of HDP, PROM, V/ASD and hyperbilirubinemia for those women with SCH who did not attain sustained euthyroid status compared to normal pregnancies. The temporal correlation between the diagnosis of SCH (LT4 treatment) in different trimesters, and pregnancy outcomes varied. Since GDM is diagnosed in the second trimester and hypertensive disorders of pregnancy also

**Table 2  Comparison of pregnancy outcomes between SCH and control groups.**

| | Outcomes | | SCH group | Non-SCH group | $p$ |
|---|---|---|---|---|---|
| Pregnant women | | | $n = 266$ | $n = 672$ | |
| | GDM | No | 230 (86.50) | 582 (86.60) | 0.954 |
| | | Yes | 36 (13.50) | 90 (13.40) | |
| | HDP | No | 251 (94.40) | 654 (97.30) | 0.027 |
| | | Yes | 15 (5.60) | 18 (2.70) | |
| | PROM | No | 180 (67.70) | 501 (74.60) | 0.033 |
| | | Yes | 86 (32.30) | 171 (25.40) | |
| Newborn | | | | | |
| | V/ASD | No | 255 (95.90) | 660 (98.20) | 0.036 |
| | | Yes | 11 (4.10) | 12 (1.80) | |
| | RDS | No | 259 (97.40) | 665 (99.00) | 0.070 |
| | | Yes | 7 (2.60) | 7 (1.00) | |
| | PTB | No | 255 (95.90) | 658 (97.90) | 0.079 |
| | | Yes | 11 (4.10) | 14 (2.10) | |
| | AWB | Low | 249 (93.60) | 637 (94.80) | |
| | | Normal | 9 (3.40) | 11 (1.60) | 0.231 |
| | | High | 8 (3.00) | 24 (3.60) | |
| | Hyperbilirubinemia | No | 155 (58.30) | 450 (67.00) | 0.012 |
| | | Yes | 111 (41.70) | 222 (33.00) | |
| | Pneumonia | No | 250 (94.00) | 657 (97.80) | 0.003 |
| | | Yes | 16 (6.0) | 15 (2.20) | |

**Notes.**
The SCH group included 219 women with SCH who were cured and 47 women with SCH during pregnancy who were not cured.

tend to occur in the second and third trimesters, the present study was only able to detect an association of GDM with SCH in the first trimester.

A meta-analysis had shown that SCH in pregnancy was associated with an increased risk of HDP regardless of the stage of the pregnancy (*Han et al., 2022*). However, the present study found that the increased risk of HDP was only associated with SCH in the first and second trimesters of pregnancy, but not in the third trimester. This may be related to the subsequent treatment after the SCH is detected.

Thyroid hormones have cardiovascular regulatory effects and chronic thyroid hormone disorders can lead to cardiovascular dysfunction (*Fletcher & Weetman, 1998*; *Danzi & Klein, 2012*; *Sheffield & Cunningham, 2004*; *Danzi & Klein, 2003*; *Rodondi et al., 2005*). A molecular study has also shown that patients with SCH have reduced nitric oxide secretion and impaired endothelium-associated vasodilatation (*Taddei et al., 2003*), which leads to increased blood pressure.

A 2019 meta-analysis on thyroid function in pregnancy and PTB showed that SCH was significantly associated with a higher risk of PTB (*Korevaar et al., 2019*). In this study, we found that compared with non-SCH pregnant women, although SCH pregnant women were treated with LT4, the incidence of PTB in both the second and third trimesters of

**Table 3 Comparison of pregnancy outcomes between the SCH and Non-SCH groups at different stages of pregnancy.**

| | Pregnancy outcomes | | SCH group | Non-SCH group | $p$ |
|---|---|---|---|---|---|
| First trimester | | | $n = 125$ | $n = 813$ | |
| | GDM | No | 99 (79.20) | 713 (87.70) | 0.009 |
| | | Yes | 26 (20.80) | 100 (12.30) | |
| | HDP | No | 113 (90.40) | 792 (97.40) | 0.001 |
| | | Yes | 12 (9.60) | 21 (2.60) | |
| | Pneumonia | No | 116 (92.80) | 791 (97.30) | 0.019 |
| | | Yes | 9 (7.20) | 22 (2.70) | |
| Second trimester | | | $n = 77$ | $n = 861$ | |
| | HDP | No | 71 (92.20) | 834 (96.90) | 0.046 |
| | | Yes | 6 (7.80) | 27 (3.10) | |
| | PTB | No | 71 (92.20) | 842 (97.80) | 0.011 |
| | | Yes | 6 (7.80) | 19 (2.20) | |
| | AWB | Normal | 71 (92.20) | 815 (94.70) | |
| | | Low | 6 (7.80) | 14 (1.60) | 0.002 |
| | | High | 0 (0.00) | 32 (3.70) | |
| | Hyperbilirubinemia | No | 36 (46.80) | 569 (66.10) | 0.001 |
| | | Yes | 41 (53.20) | 292 (33.90) | |
| | RDS | No | 73 (94.80) | 851 (98.80) | 0.021 |
| | | Yes | 4 (5.20) | 10 (1.20) | |
| Third trimester | | | $n = 119$ | $n = 819$ | |
| | PTB | No | 111 (93.30) | 802 (97.90) | 0.008 |
| | | Yes | 8 (6.70) | 17 (2.10) | |

**Notes.**
The research subjects of the SCH group are only the SCH affected population in this pregnancy stage. There are a total of 321 people in the SCH groups of three pregnancy stages, of which 47 pregnant women were included twice and eight pregnant women were included three times. The research subjects of the non-SCH group are the non SCH affected population in this pregnancy stage.

pregnancy was higher. Preterm infants are often associated with a range of complications such as low birth weight and RDS.

In this study, 47 (5.01) pregnant women had suboptimally treated SCH in at least two trimesters. A comparison of the STS, SES and non-SCH groups showed that the STS group had a higher risk of HDP, PROM, PTB, AWB, hyperbilirubinemia, V/ASD and RDS than the non-SCH group. There was no statistical difference between the SES and non-SCH groups. This suggests that LT4 therapy, which keeps maternal thyroid function at normal levels throughout the pregnancy cycle, is effective in reducing the risk of pregnancy-related adverse events. In compiling the data, we also found the phenomenon that LT4 therapy was highly effective, with normal thyroid function test data once or twice after dosing. However, after two to three months, SCH is prone to recurrence for the remainder of the current pregnancy, due to irregular medication or maternal discontinuation. This also reminds us that multiple thyroid function tests during pregnancy are necessary to detect abnormal thyroid function, to treat with LT4 in a timely manner, especially in pregnant women who already have abnormal thyroid function. The results of correlation of TSH levels at

**Table 4** A multifactorial logistic regression model of SCH in pregnancy and five pregnancy outcomes.

| Outcomes | | *p* | OR | 95% CI |
|---|---|---|---|---|
| HDP | Model 1 | 0.029 | 2.204 | 1.083–4.487 |
| | Model 2 | 0.035 | 3.201 | 1.087–9.423 |
| | Model 3 | 0.066 | 2.822 | 0.932–8.543 |
| PROM | Model 1 | 0.031 | 1.412 | 1.032–1.932 |
| | Model 2 | 0.130 | 1.378 | 0.910–2.085 |
| | Model 3 | 0.137 | 1.373 | 0.904–2.086 |
| V/ASD | Model 1 | 0.030 | 2.589 | 1.098–6.105 |
| | Model 2 | 0.073 | 2.809 | 0.910–8.677 |
| | Model 3 | 0.060 | 2.989 | 0.953–9.378 |
| Hyperbilirubinemia | Model 1 | 0.008 | 1.486 | 1.107–1.995 |
| | Model 2 | 0.001 | 2.128 | 1.435–3.154 |
| | Model 3 | 0.001 | 2.114 | 1.421–3.144 |
| Pneumonia | Model 1 | 0.003 | 3.018 | 1.447–6.295 |
| | Model 2 | 0.003 | 3.816 | 1.597–9.119 |
| | Model 3 | 0.002 | 3.975 | 1.655–9.551 |

Notes.

Model 1 adjusted for age, BMI, parity, smoking and alcohol consumption, Model 2 added education, family income, family history of diabetes and history of induced abortion to Model 1, and Model 3 added folic acid, VD supplementation to Model 2.

**Table 5** Comparison of pregnancy outcomes between the cured, non-cured and control groups.

| Outcomes | | Non-SCH group $n = 672$ | SES group $n = 219$ | STS group $n = 47$ | *p* |
|---|---|---|---|---|---|
| HDP | No | 654 (97.30)[a] | 211 (96.30)[a] | 40 (85.10)[b] | 0.001 |
| | Yes | 18 (2.70)[a] | 8 (3.70)[a] | 7 (14.90)[b] | |
| PROM | No | 501 (74.60)[a] | 152 (69.40)[ab] | 28 (59.60)[b] | 0.040 |
| | Yes | 171 (25.40)[a] | 67 (30.60)[ab] | 19 (40.40)[b] | |
| PTB | No | 658 (97.90)[a] | 214 (97.70)[a] | 41 (87.20)[b] | 0.002 |
| | Yes | 14 (2.10)[a] | 5 (2.30)[a] | 6 (12.80)[b] | |
| AWB | Normal | 637 (94.80)[a] | 207 (94.50)[a] | 42 (89.40)[b] | |
| | Low | 11 (1.60)[a] | 4 (1.80)[a] | 5 (10.60)[b] | 0.015 |
| | High | 24 (3.60)[a] | 8 (3.70)[a] | 0 (0.00)[b] | |
| Hyperbilirubinemia | No | 450 (67.00)[a] | 133 (60.70)[ab] | 22 (46.80)[b] | 0.008 |
| | Yes | 222 (33.00)[a] | 86 (39.30)[ab] | 25 (53.20)[b] | |
| V/ASD | No | 660 (98.20)[a] | 212 (96.80)[ab] | 43 (91.50)[b] | 0.043 |
| | Yes | 12 (1.80)[a] | 7 (3.20)[ab] | 4 (8.50)[b] | |
| RDS | No | 665 (99.0)[a] | 215 (98.20)[ab] | 44 (93.60)[b] | 0.027 |
| | Yes | 7 (1.00)[a] | 4 (1.80)[ab] | 3 (6.40)[b] | |

Notes.

The Bonferroni method is used to evaluate $\alpha$ Perform correction, and differences in a and b between groups indicate statistically significant differences.

different stages of pregnancy showed a tendency to increase with increasing gestational weeks, and a recent retrospective study showed similar results (*Fan et al., 2013*). Therefore, thyroid function monitoring during pregnancy and timely intervention is necessary to

**Table 6   Results of Spearman's rank correlation of TSH levels in different trimesters.**

| Total population $n = 938$ | Gestation stage | $r$ | $p$ |
|---|---|---|---|
| | First and second trimester | 0.114 | 0.001 |
| | First and third trimester | 0.086 | 0.008 |
| | Second and third trimester | 0.550 | 0.001 |
| Non-SCH group $n = 672$ | First and second trimester | 0.179 | 0.001 |
| | First and third trimester | 0.208 | 0.001 |
| | Second and third trimester | 0.573 | 0.001 |

treat pregnant women whose TSH levels have exceeded or are close to the upper limit of the reference value.

In contrast to other studies, this study focused on the dynamics of thyroid function throughout pregnancy in the study population, particularly in the period after LT4 treatment was initiated. LT4 treatment was able to correct the state of abnormal thyroid function, affirming the effectiveness of LT4 treatment for SCH. The results of the subgroup comparison, which varied according to the therapeutic non-SCH of SCH, showed that effective LT4 treatment, which resulted in SCH not recurring for the remainder of the current pregnancy, was beneficial for maternal and neonatal pregnancy outcomes. Of course, as a single-centre study involving only pregnant women who came to Shijiazhuang Obstetrics and Gynecology Hospital for examinations and deliveries, there may be subject to selection bias. Also, the sample size was small compared to other large cohort studies. Larger prospective studies are needed to validate our conclusions.

## CONCLUSIONS

Therefore, thyroid function screening and monitoring are recommended for women preparing for pregnancy and during pregnancy. For pregnant women who are diagnosed with SCH before pregnancy or early in pregnancy, timely initiation of LT4 therapy to attain and sustain euthyroid status for the entire pregnancy will benefit both the mother and newborn.

## ACKNOWLEDGEMENTS

We would thank patients and their family for this research understanding.

### Funding
The authors received no funding for this work.

### Competing Interests
The authors declare there are no competing interests.

## Author Contributions

- Yutian Zhou conceived and designed the experiments, performed the experiments, analyzed the data, prepared figures and/or tables, authored or reviewed drafts of the article, and approved the final draft.
- Yi Wang conceived and designed the experiments, performed the experiments, analyzed the data, prepared figures and/or tables, and approved the final draft.
- Tianxiao Yu conceived and designed the experiments, performed the experiments, analyzed the data, prepared figures and/or tables, authored or reviewed drafts of the article, and approved the final draft.
- Yuan Li performed the experiments, analyzed the data, prepared figures and/or tables, and approved the final draft.
- Meiyan Mi performed the experiments, prepared figures and/or tables, and approved the final draft.
- Jianqiang Su performed the experiments, prepared figures and/or tables, and approved the final draft.
- Jun Ge conceived and designed the experiments, prepared figures and/or tables, authored or reviewed drafts of the article, and approved the final draft.

## Human Ethics

The following information was supplied relating to ethical approvals (*i.e.*, approving body and any reference numbers):

The Ethics Committee of The Fourth Hospital of Shijiazhuang (No: 20230109) approved the study. All the participants were informed about the study and a signed permission document was obtained in accordance with the Declaration of Helsinki.

## Clinical Trial Ethics

The following information was supplied relating to ethical approvals (i.e., approving body and any reference numbers):

The Ethics Committee of The Fourth Hospital of Shijiazhuang approved the study (No: 20230109).

## Data Availability

The raw data are available in the Supplemental File.

## Clinical Trial Registration

The following information was supplied regarding Clinical Trial registration:

20230109

## Supplemental Information

Supplemental information for this article can be found online at http://dx.doi.org/10.7717/peerj.19343#supplemental-information.

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
