# Peer review of "Subclinical hypothyroidism during pregnancy and the impact of levothyroxine therapy on pregnancy outcomes in women"

_PeerJ, doi:10.7717/peerj.19343_

## Round 0.1 · original submission · Major Revisions

Although the work has been appreciated, to improve their manuscript, the authors should address all the issues raised by the reviewers.

·

Basic reporting

Clear and unambiguous, professional English has been used throughout the manuscript.
In the line no. 36, the word "with" should be made capital.
In the line no. 58 please use singular word for "unit".

Experimental design

Yes

Validity of the findings

Yes

Additional comments

The references are not in the Vancouver style, please re-arrange it in the Vancouver style.

·

Basic reporting

Professional English is used throughout but there are certain terms that do not precisely capture their intended meaning. I detail these in the attached remarked PDF of the manuscript.
The literature references appear to be pertinent to the points in the text for their cited. I am familiar with some of these references and for those in particular, I have high confidence that they are being properly utilized.
The article was constructed in a proper professional manner. The figures are simple and clear. The tables have all of the essential information but could be reformatted to ease the reader's ability to comprehend. I did not see a means for accessing the raw data; however, I would be unable to invest the time and effort to review all of the raw data were it at hand.
The aims of the study and the outcomes are well matched, so whatever is meant by "self-contained" I believe is fulfilled.

Experimental design

This is original research derived from clinical data from the labor and delivery service of their hospital.
The research question is well-defined, relevant to obstetrical and neonatal outcomes and is important for filling a gap relating to the inadequacies of preceding studies in the literature wherein assessment for maternal subclinical hypothyroidism is either inconsistently performed or conducted late enough into an established pregnancy to render some of the past reported outcomes questionable.
It appears the current investigation was performed with high technical and ethical standards.
The methods are mostly described with sufficient detail and information to replicate, but a few specific questions about exclusion and inclusion criteria are presented in the remarked PDF.

Validity of the findings

The rationale for performing this work and the ability of the study to add to the literature is clearly stated.
As noted above, I did not seek the raw data and would be unable to invest the time and effort to confirm whether the raw data are reflective of the summary statistics presented in the manuscript. The statistical methods appear appropriate to me; however, I do advise the Journal seek the input from a biostatistician regarding the modeling work and its interpretation.
While there are numerous elements of the text that are remarked in the PDF that need attention, ultimately the discussion does present justifiable points that are important and are well-made by the authors.

Additional comments

My primary additional comment is that the authors need to consider the numerous suggestions I have included in the remarked PDF. Their study seems to be well conceived but a number of important aspects need to be elaborated or clarified in order for confidence in the study conduct, outcomes and interpretation to be strengthened enough to permit future publication.

Reviewer 3 ·

Basic reporting

Thanks to the Editor for this opportunity to review the manuscript entitled “Subclinical hypothyroidism (SCH) during pregnancy and the impact of LT4 therapy on pregnancy outcomes in women”. The manuscript reports the results of a single-centre observational and retrospective study on the impact of LT4 therapy on pregnancy outcomes of two groups: 266 women with SCH (219 cured and 47 non-cured) and 672 control women. The main results are that women with SCH presented on the maternal significantly side more hypertensive disorders of pregnancy/HDP (p=0.027) and premature rupture of membranes/PROM (p=0.033) and on the neonatal side significantly more neonatal ventricular or atrial septal defect/V/ASD (p= 0.036), hyperbilirubinemia (p=0.012) and pneumonia (p= 0.003). This manuscript deals with a crucial point in thyroidology since evidence linking SCH and an eventual LT4 treatment to adverse pregnancy outcomes is inconsistent and conflicting. The authors stated it well in their introduction. This work is also of interest because it reports data from a specific region (East of China).
Basic reporting:
The language is clear throughout the manuscript, a minor editing may be pertinent. The global structure is correct. Tables and figures are clear but Authors could considerer sending some tables as supplemental data. Please be aware that Table reported the comparison between the two groups for First trimester only and not the two others as sated in the text (lines 136-139). Unidentified raw data are available but it seems that further information and metadata are missing to exploit them correctly. Regarding the literature, Authors cited the meta-analysis and systematic review of Maraka et al (2016). They should update with that of Sankoda et al. (2024 PMID 38368537). One of the main issues of the study lies in the criteria used to initiate LT4 treatment for SCH. Authors stated line 56 that “LT4 therapy is currently commonly used to treat SCH during pregnancy” citing references 14 (2012) and 15 (2019). That is not quite right. According to ATA 2017 (the most cited recommendations and reference 13 of the manuscript), it is recommended to treat or to considerer treating SCH with LT4 depending on the level of TSH and the presence of anti-TPO antibodies. Authors seemed to follow Chinese recommendations (ref 20), it would be interesting to develop and to compare them to those of ATA (at least in the discussion section) and to provide some context data such as the iodine intake and thyroid diseases prevalence.

Experimental design

The second issue is methodological. Regarding the lack of evidence, the literature (like Sankoda et al.) recommends further well-designed studies. It is obvious that a single-center retrospective and observational study will not contribute efficiently to the debate.

Validity of the findings

A third major issue is the reported baseline characteristics of the patients. Why TSH levels and anti-TPO antibodies are not reported in Table 1? These variables are of upmost importance for the study. How did authors distinguish cured from non-cured SCH? The added-value of Table 4 (multifactorial logistic regression) and Figure 2 (correlation of TSH levels in different trimesters) is not obvious in this setting.

---

## Round 0.2 · accepted · Accept

The authors have adequately revised their manuscript by addressing the concerns raised by the reviewers , so that the work may be now accepted for publication.

Reviewer 3 ·

Basic reporting

Improved.

Experimental design

No comment.

Validity of the findings

The conceptual change from cured/non-cured to SES/STS is pertinent.

Additional comments

The Authors answered the questions and followed most of the Reviewers' recommendations.